# A polar coordinate system represents syntax in large language models

**Pablo Diego-Simón**
ENS, PSL University, Paris, France
pablo-diego.simon@psl.eu

**Stéphane D'Ascoli**
Meta AI, Paris, France
stephane.dascoli@gmail.com

**Emmanuel Chemla**
ENS, PSL University, Paris, France
emmanuel.chemla@ens.psl.eu

**Yair Lakretz**
ENS, PSL University, Paris, France
yair.lakretz@gmail.com

**Jean-Rémi King**
Meta AI, Paris, France
jeanremi@meta.com

## Abstract

Originally formalized with symbolic representations, syntactic trees may also be effectively represented in the activations of large language models (LLMs). Indeed, a "Structural Probe" can find a subspace of neural activations, where syntactically-related words are relatively close to one-another. However, this syntactic code remains incomplete: the distance between the Structural Probe word embeddings can represent the *existence* but not the *type* and *direction* of syntactic relations. Here, we hypothesize that syntactic relations are, in fact, coded by the relative direction between nearby embeddings. To test this hypothesis, we introduce a "Polar Probe" trained to read syntactic relations from both the distance and the direction between word embeddings. Our approach reveals three main findings. First, our Polar Probe successfully recovers the type and direction of syntactic relations, and substantially outperforms the Structural Probe by nearly two folds. Second, we confirm that this polar coordinate system exists in a low-dimensional subspace of the intermediate layers of many LLMs and becomes increasingly precise in the latest frontier models. Third, we demonstrate with a new benchmark that similar syntactic relations are coded similarly across the nested levels of syntactic trees. Overall, this work shows that LLMs spontaneously learn a geometry of neural activations that explicitly represents the main symbolic structures of linguistic theory.

## 1 Introduction

Human languages have long been proposed to systematically follow tree-like structures (Chomsky, 1957; Tesnière, 1953). In a sentence, words that are far apart can be syntactically linked. For example "cats" is the subject of "chase" in the sentence "The cats in cities chase the mice". In dependency grammar, the edges of such trees are directed and labelled to indicate the type of syntactic relation between words ("cats" is the subject of "chase", Fig. 1B).

Despite their conceptual soundness and alignment with human behavior (Robins, 2013), syntactic trees have long been the crux of a core challenge in cognitive science (Smolensky, 1987): trees are symbolic representations, which can superficially appear incompatible with the vectorial representations of neural networks. This opposition between symbols and vectors has been a major challenge to the

38th Conference on Neural Information Processing Systems (NeurIPS 2024).

unification of linguistic theories on the one hand, and neuroscience and connectionist AI on the other hand.

Recently, Hewitt and Manning (Hewitt and Manning, 2019) proposed an important concept for this issue, by suggesting that the existence of syntactic link between two words may be represented by the distance between their corresponding embeddings. Specifically, their "Structural Probe" consists in finding a subspace of contextualized word embeddings such that the squared euclidean distance between words represents their distance in the dependency tree. They showed that the Structural Probe is most powerful in the intermediate layers of language models: these layers contain a subspace where syntactically-related words are closer together.

This Structural Probe, however, can only reveal one aspect of dependency trees: namely, the *existence* of syntactic relations, between word pairs. However, whether and how the *direction* and the *type* of syntactic relations are represented in language models remains unknown.

Here, we hypothesize that syntactic relations are represented by a polar coordinate system, where the *existence* and *type* of syntactic relations are represented by *distances* and *direction*, respectively (Fig. 1). To test this hypothesis, we introduce a "Polar Probe": a linear transformation trained such that pairs of words linked by the same dependency type are collinear, while remaining orthogonal to different dependency types.

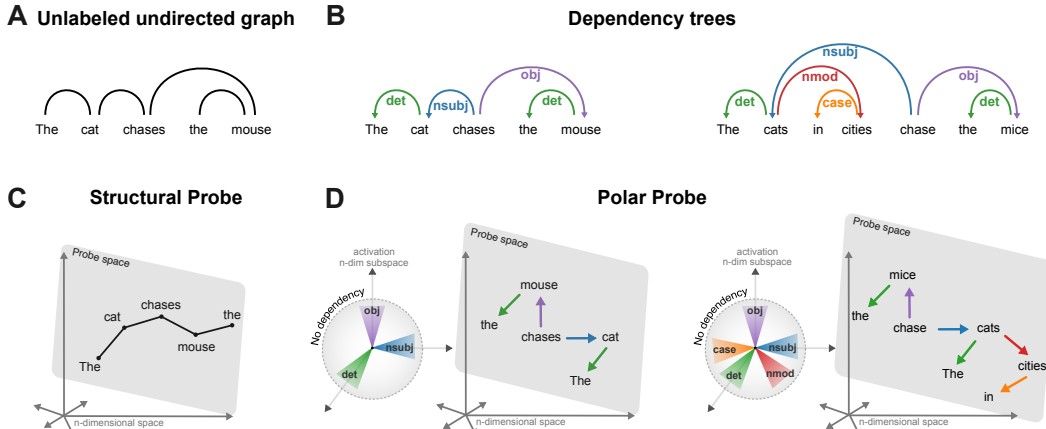

Figure 1: **Dependency trees hypothesized in linguistics and in neural networks. A.** According to the dependency grammar framework, the sentences can be described as linear sequences of words connected by an acyclic graph. **B.** More precisely, such acyclic graph is both labeled and directed, where each edge has a direction, representing the hierarchy of the syntactic relation, and a label, representing the type of syntactic relation. **C.** The Structural Probe (Hewitt and Manning, 2019) finds a a linear transform (gray plane) of the language model's activations (here simplified as a 3D space), such that the distance between word embeddings is predicted by their dependency tree. In the Structural Probe subspace, however, it is not possible to distinguish whether "The cat chases the mouse" or "The mouse chases the cat." **D.** Our Polar Probe finds a linear transformation where the angle between syntactically-related word additionally represents the type and direction of these syntactic relations, and the distance codes its presence. The colored arrows indicate *orthogonal* directions in the Polar-Probe subspace.

## 2 Methods

The goal of this work is to find a linear readout of the activations of pretrained language models which explicitly represents both the presence and the types of syntactic relations between words.

### 2.1 Problem statement

Dependency grammar represents the syntax of a sentence as a symbolic tree. Accordingly, let:

- $w_i$ be the $i^{th}$ word in a sentence,
- $d : w_i, w_j \mapsto \mathbb{Z}^+$ indicate the syntactic distance between two words,

- $C$ be the set of syntactic types,
- $t : w_i, w_j \mapsto C$ indicate the type of syntactic relation between directly-connected words.
- $u : w_i, w_j \mapsto \{w_i, w_j\}$ indicate the head word (and thus direction) of the syntactic relation between directly-connected words.

Language models based on neural networks represent sentences as sequences of word vectors (a.k.a word embeddings)[1]. As these embeddings propagate through the layers of the network, they incorporate information about the sentence they belong within, becoming *contextualized* word embeddings.

Let $\mathbf{h}_i^\ell \in \mathbb{R}^k$ be the $i^{th}$ contextualized word embedding of $w_i$ obtained at the output of layer $\ell$ of a neural network. For simplicity, we will drop the layer index $\ell$ in what follows, keeping in mind that different layers yield different representations.

Retrieving syntactic trees from a neural network requires to obtain three read-out functions $\hat{d} : \mathbb{R}^k, \mathbb{R}^k \mapsto \mathbb{Z}^+$, $\hat{u} : \mathbb{R}^k, \mathbb{R}^k \to \mathbb{R}^k$ and $\hat{t} : \mathbb{R}^k, \mathbb{R}^k \mapsto C$ such that the following conditions are met:

$$\hat{d}(\mathbf{h}_i, \mathbf{h}_j) \approx d(w_i, w_j), \qquad \hat{t}(\mathbf{h}_i, \mathbf{h}_j) \approx t(w_i, w_j), \qquad \hat{u}(\mathbf{h}_i, \mathbf{h}_j) \approx u(w_i, w_j),$$

While $\hat{d}$, $\hat{t}$ and $\hat{u}$ could be any complex functions, the goal of the present work is to identify a simple, interpretable "code" of how syntactic trees may be represented in vectorial systems. Following the classic definition of a representation as a linearly readable information, we focus on linear operators (DiCarlo and Cox, 2007; Kriegeskorte and Bandettini, 2007; King and Dehaene, 2014).

## 2.2 Structural Probe

In (Hewitt and Manning, 2019), the authors propose to solve $\hat{d}$ as a distance in a subspace of the contextualized word embeddings. Their "Structural Probe" is a linear transform, $\mathbf{B}_S : \mathbb{R}^k \mapsto \mathbb{R}^{k\prime}$, which projects word embeddings such that their relative distances correspond to their distances in the syntactic tree. Formally, if $\mathbf{s}_{i,j} \in \mathbb{R}^k$ is the (directed) "edge embedding" between words $w_i$ and $w_j$:

$$\mathbf{s}_{i,j} = \mathbf{h}_i - \mathbf{h}_j \in \mathbb{R}^k, \tag{1}$$

Then, the predicted distance $\hat{d} \in \mathbb{R}^+$ between two words can be computed directly as a function of this and no other information coming from the individual word embeddings[2] :

$$\hat{d}(\mathbf{h}_i, \mathbf{h}_j) := \|\mathbf{B}_S \mathbf{s}_{ij}\|^2. \tag{2}$$

Given a set of sentences, the authors extract the set $\Omega_S$ of pairs of words belonging to the same sentence and optimize $\mathbf{B}_S$ to minimize the absolute difference between the distances within the syntactic tree and the distances between the probed word embeddings:

$$\mathcal{L}_S(\mathbf{B}_S) = \frac{1}{|\Omega_S|} \sum_{(w_i, w_j) \in \Omega_S} |d(w_i, w_j) - \hat{d}(w_i, w_j)| \tag{3}$$

Following the development of the Structural Probe, squared Euclidean distances between probed word embeddings are not designed to represent both the presence of dependency relations and their types and directions simultaneously. (Hewitt and Manning, 2019) thus only propose a representational system to solve $\hat{d}$, but not $\hat{t}$ and $\hat{u}$. Whether and how the directed and labeled syntactic tree is encoded in neural networks, thus remains unknown.

---

[1] Sequences are built from "tokens", which sometimes correspond to subwords. When this is the case, one can simply average the subword embeddings to obtain word embeddings (Hewitt and Manning, 2019).

[2] See (Chen et al., 2021) for an explanation of how hyperbolic spaces are best measured through *square* Euclidean distances.

## 2.3 Angular Probe

Here, we hypothesize that neural networks use the *orientation* of the relations formed by connected word pairs to represent the type and direction of their syntactic dependency.

To test this hypothesis, we first introduce an "Angular Probe" consisting of a linear transform $\mathbf{B}_A : \mathbb{R}^{k'} \mapsto \mathbb{R}^{k''}$. By abuse of notation, we denote as $t(\mathbf{s}_{i,j}) \equiv t(w_i, w_j)$ the syntactic type of the corresponding edge. For the Structural Probe, the function $d$ to be recovered is defined on all pairs of words; here the function $t$ to be recovered is only defined on pairs of syntactically linked words, hence we only consider word pairs $(w_i, w_j)$ which are indeed syntactically linked.

We use contrastive learning to align relations of the same type and ensure that different types are pointing to different directions. This approach is designed such that a linear readout could explicitly categorize dependency types.

Specifically, the objective for the Angular Probe is to ensure that given two edge embeddings $\mathbf{s}$ and $\mathbf{s}'$ of syntactic types $c = t(\mathbf{s})$ and $c' = t(\mathbf{s}')$, the linear transforms $\mathbf{B}_A \mathbf{s}$ and $\mathbf{B}_A \mathbf{s}'$ are colinear if $c = c'$, and orthogonal if $c \neq c'$.

Formally, the Angular Probe is the linear transform which minimizes the contrastive loss:

$$\mathcal{L}_A(\mathbf{B}_A) = \frac{1}{\Omega_A} \sum_{s,s' \in \Omega_A} \left( \measuredangle(\mathbf{B}_A \mathbf{s}, \mathbf{B}_A \mathbf{s}') - \mathbb{1}\left[t(\mathbf{s}) = t(\mathbf{s}')\right] \right)^2, \tag{4}$$

- $\Omega_A$ is the set of edge embeddings of syntactically connected words,
- $\measuredangle : \mathbf{x}, \mathbf{y} \mapsto \frac{\mathbf{x} \cdot \mathbf{y}}{\|\mathbf{x}\|\|\mathbf{y}\|}$ is the cosine similarity,
- $\mathbb{1} : \mathcal{X} \mapsto \begin{cases} 1 \text{ if } \mathcal{X} \text{ is true} \\ 0 \text{ otherwise} \end{cases}$ is the indicator function.

We can then construct a prototypical vector for each dependency type, by averaging all the probed edge embeddings belonging to that type:

$$\mathbf{V}_c = \sum_{\mathbf{s} \in \Omega_A^{(c)}} \mathbf{B}_A \mathbf{s}, \quad \Omega_A^{(c)} = \{ \mathbf{s} \in \Omega_A | t(\mathbf{s}) = c \}. \tag{5}$$

The Angular Probe gives us the following function to retrieve the syntactic type of any given edge:

$$\hat{t}(\mathbf{h}_i, \mathbf{h}_j) := \underset{c}{\operatorname{argmax}} \left| \measuredangle(\mathbf{B}_A \mathbf{s}_{ij}, \mathbf{V}_c) \right|. \tag{6}$$

We also get the function $\hat{u}$ to predict the head word (and direction) of the syntactic relation given a predicted type $\hat{t}$:

$$\hat{u}(\mathbf{h}_i, \mathbf{h}_j) := \begin{cases} \mathbf{h}_i & \text{if } \measuredangle(\mathbf{B}_A \mathbf{s}_{ij}, \mathbf{V}_{\hat{t}(\mathbf{h}_i, \mathbf{h}_j)}) \geq 0 \\ \mathbf{h}_j & \text{if } \measuredangle(\mathbf{B}_A \mathbf{s}_{ij}, \mathbf{V}_{\hat{t}(\mathbf{h}_i, \mathbf{h}_j)}) < 0 \end{cases} \tag{7}$$

## 2.4 Polar Probe

The Angular Probe and the Structural Probe have independent objectives applied to two different datasets: $\mathcal{L}_S$ relies on $\Omega_S$, which includes all word pairs from any sentence, whereas $\mathcal{L}_A$ relies on $\Omega_A$, which contains only pairs of words that are syntactically linked.

Consequently, we define the Polar Probe as a single linear transformation $\mathbf{B}_P : \mathbb{R}^k \mapsto \mathbb{R}^{k''}$ which results from the joint optimization of both the the Angular and Structural objectives.

This Polar Probe defines the final functions to identify syntactic distances and relation types:

$$\hat{d}(\mathbf{h}_i, \mathbf{h}_j) := \|\mathbf{B}_P \mathbf{s}_{ij}\|^2 \tag{8}$$

$$\hat{t}(\mathbf{h}_i, \mathbf{h}_j) := \underset{c}{\operatorname{argmax}} |\angle(\mathbf{B}_P \mathbf{s}_{ij}, \mathbf{V}_c)| \tag{9}$$

$$\hat{u}(\mathbf{h}_i, \mathbf{h}_j) := \begin{cases} \mathbf{h}_i & \text{if } \angle(\mathbf{B}_P \mathbf{s}_{ij}, \mathbf{V}_{\hat{t}(\mathbf{h}_i, \mathbf{h}_j)}) \geq 0 \\ \mathbf{h}_j & \text{if } \angle(\mathbf{B}_P \mathbf{s}_{ij}, \mathbf{V}_{\hat{t}(\mathbf{h}_i, \mathbf{h}_j)}) < 0 \end{cases} \tag{10}$$

Therefore, the Polar Probe minimizes the following loss function, with the hyper-parameter $\lambda$ weighing the Angular objective.

$$\underset{\mathbf{B}_P}{\operatorname{argmin}} \, \mathcal{L}_S(\mathbf{B}_P) + \lambda \mathcal{L}_A(\mathbf{B}_P) \tag{11}$$

## 2.5 Data

**Natural dataset.** We consider natural sentences extracted from the English Web Treebank dataset (Silveira et al., 2014)[3]. This corpus contains 254,820 words from 16,622 sentences, sourced from a diverse array of web media genres, including weblogs, newsgroups, emails or reviews. All the sentences in the dataset are manually annotated according to the Universal Dependencies framework (Nivre et al., 2017), where each word is a node, each syntactic link is a labelled directed edge, and the syntactic tree is acyclic.

Sentences containing email or web addresses are excluded from the dataset. Such filter removes noisy sentences not interesting from a syntactic point of view. We follow the default splitting provided by the English Web TreeBank resulting in a total of 11827 sentences for training, 1851 for validation, and 1869 for testing.

**Controlled dataset.** To precisely evaluate our approach on well-controlled sentences, we designed a dataset, extending previous work (Lakretz et al., 2021b), comprising 100 sentences built with a long-nested structure (e.g., "*The book that the boy besides the car reads fascinates my teacher*"). In these sentences, a subordinate clause branches off the main phrase. There is also a constituent '*besides the car*' that forms a branch inside the subordinate clause, adding further complexity to the syntactic structure.

This controlled dataset is designed to clarify how the Polar Probe reconstructs the syntactic tree in complex conditions, but conditions well theorized in linguistics. In particular, we can create several variations of these long-nested sentences.

- Short: "*The book fascinates my teacher*"
- Relative clause: "*The book that the boy reads fascinates my teacher*"
- Long-nested: "*The book that the boy besides the car reads fascinates my teacher*"

Thanks to this dataset we can study whether the Polar Probe maps embeddings of different syntactically-identical sentences to consistent latent locations and orientations. In addition, with the different "sentence levels", word position and syntactic role can be disentangled to compare the Polar Probe's activations accross levels, as shown in Fig: 5.

## 2.6 Training

We train the Polar Probe on the neural activations of Mistral-7B-v0.1 and Llama-2-7b-hf (Touvron et al., 2023; Jiang et al., 2023), in response to sentences of the "Natural Dataset" described above. Both models are Auto-Regressive Language Models, they aim to identify future words from input sentences. Furthermore, these models read all the words simultaneously, building representations which depend on the whole sequence of tokens.

We trained the Polar Probes with gradient descent, using the Adam optimizer (Kingma and Ba, 2014) with a learning rate of 0.005, and a batch size of 200 sentences. The duration of the training is 30 epochs, we perform model selection using the validation set. Hyperparameter $\lambda$ is set to 10.0, ensuring an optimal balance between the Angular and Structural objective.

---

[3]`https://universaldependencies.org/treebanks/en_ewt/index.html`

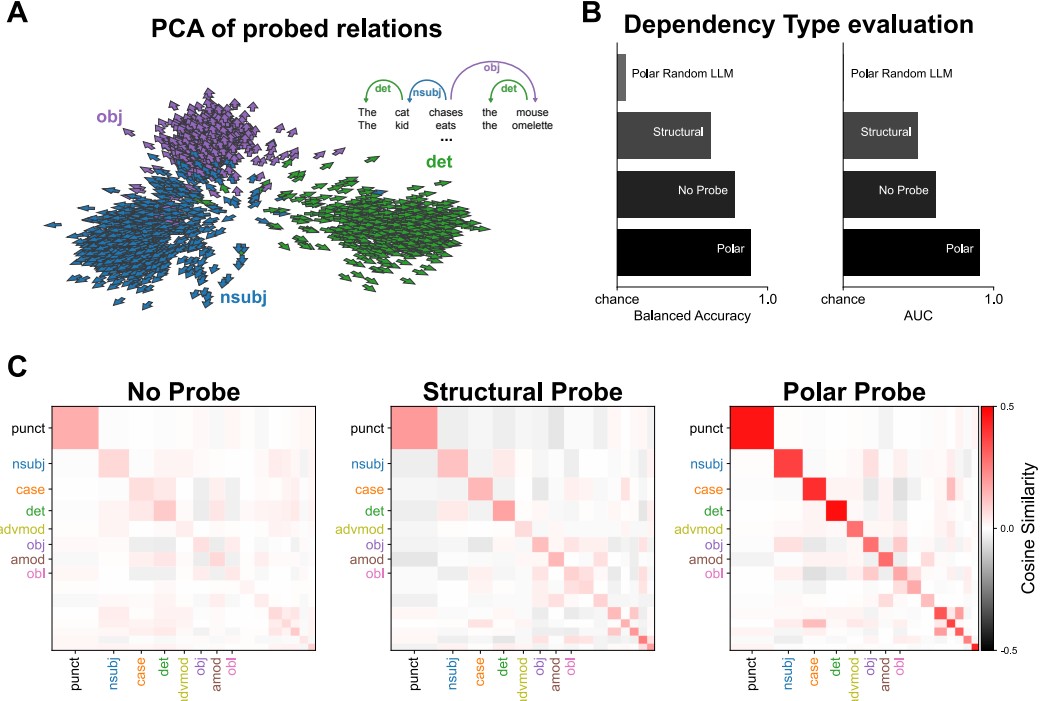

Figure 2: **The Polar Probe reliably identifies dependency types. A.** PCA visualization of edges linearly read by the Polar Probe. The color of each edge corresponds to one of three different dependency types ('nsubj', 'obj', 'det'): the linear readouts point in systematic directions. **B.** AUC and Balanced Accuracy metrics obtained for dependency type classification. **C.** Pairwise cosine similarity (0=orthogonal, 1=collinear) matrices obtained without a probe (left) the Structural Probe (middle) and the Polar Probe (right).

## 2.7 Evaluation

We evaluate each probe either on its ability to faithfully represent (i) the unlabelled and undirected dependency tree ("structure"), (ii) the type and direction of dependencies and (iii) both of these elements.

**Dependency structure.**   Following (Hewitt and Manning, 2019), we evaluate whether the probes accurately predicts the existence of each syntactic relation by using the Undirected Unlabeled Attachment Score (UUAS). UUAS quantifies the proportion of dependency relations (directly connected words) in the dependency tree that are correctly identified by the probe, irrespective of their dependency types and direction.

**Dependency type.**   To evaluate the accuracy of the predicted dependency types, we use three distinct metrics: Area Under the Curve (AUC), Dependency Type Accuracy and Dependency Type Balanced Accuracy.

For AUC, we compute the cosine similarity between each edge $\mathbf{s}_i$ and all other edges $\mathbf{s}_j$, that are either from the same dependency type, or not. This procedure ends with a distance vector $x \in \mathbb{R}^m$ of $m$ edge pairs and a binary vector of $y \in \mathbb{1}^m$. We can finally input these two vectors into scikit-learn's `roc_auc_score` (Pedregosa et al., 2011).

For Accuracy and Balanced Accuracy, we classify dependency types by comparing relations to prototypical relations. For this, from the training set, we pool 10,000 relations and define a prototype $\tilde{\mathbf{s}}_k$ for each dependency type $k$, by computing the centroid of all relations belonging to the same type $k$. Then, we predict dependency types from the cosine similarity between each relation $\mathbf{s}_i$ and each prototype $\mathbf{V}_c$, using scikit-learn's `KNeighborsClassifier`.(Pedregosa et al., 2011). Finally, we use scikit-learn `accuracy_score` or `balanced_accuracy_score` to limit the effect of imbalance between dependency types.

**Combined dependency type and structure.**   Finally, to provide a metric which evaluates *both* dependency structures and dependency types, we compute the Labeled Attachment Score (LAS). LAS is defined as the proportion of correctly predicted labeled and directed edges in a sentence.

**Baselines.**   We compare the Polar Probe to a variety of baselines: (i) Structural Probe, (ii) "Polar Probe Random LLM": a Polar Probe trained on top of a random language model and (iii) "No Probe": the raw activations of the Language Model without any transformation.

## 3   Results

**Reliable coding of dependency types.**   We first analyze the Polar Probe on the 16[th] layer of Llama-2-7b-hf, Mistral-7B-v0.1 and BERT-large (Touvron et al., 2023; Jiang et al., 2023; Devlin et al., 2019) on the English Web Treebank (EWT) sentences (Silveira et al., 2014) annotated with dependency trees. We evaluate, on an independent test set with 10000 relations, whether pairs of words linked by similar syntactic relations point towards similar orientations in the probe's representational space (Fig. 2).

Fig. 2.A shows a Principal Component Analysis (PCA) projection of the dependency relation embeddings from the test set, once linearly read by the Polar Probe. For readability, we restrict ourselves to three of the most common types of dependencies in the dataset. As expected, the three types of dependency consistently point in different directions.

We then compute the cosine similarity between all pairs of edge embeddings probed with the Polar Probe (Fig. 2.C (right)), indeed showing that relations of the same types are collinear, while relations of different types are orthogonal. This is much clearer for the Polar Probe than for baselines (Fig. 2.C).

**Comparison with baselines.**   In Fig. 2.B, we summarize with AUC and Balanced Accuracy the extent to which the orientations of these edge embeddings reliably represent the syntactic types.

On average across dependency types, the Polar Probe reaches a AUC score of 95%, well above the Structural Probe (AUC=74%), the No Probe (AUC=80%) and the Polar Probe trained on a Randomly Initialized LLM (AUC=50%). The same relative results across probes are conserved for the Balanced Accuracy score. Importantly, these results confirm that the Polar Probe outperforms the Structural Probe in predicting dependency types from the probe embeddings. The latter is therefore something not emergent in the Structural Probe.

Unexpectedly, "No Probe" predicts syntactic types well above chance, and significantly better than the Structural. This hints to the fact that syntactic types are already represented in the raw activations. A likely explanation for this is that words belonging to the same part-of-speech (such as verbs, nouns) are clustered in the embedding space, thus partially guiding the inference of syntactic dependencies.

Moreover, training a Polar Probe on a random initialization of a language model does not accomplish the contrastive objective better than chance. Resulting in near chance-level Balanced Accuracy and AUC scores. This confirms that the linear probe needs a rich underlying representation space to work, and cannot learn to cluster the different syntactic types on its own.

**Layer-wise analysis.**   To evaluate how the Angular and Structural performance interact in the Polar Probe, and whether the mechanism generalizes to both Mistral-7B-v0.1, Llama-2-7b-hf and BERT-large, we evaluate the layers of the three models on Labeled Attachment Score (LAS) (Fig. 3). (See Supplementary for BERT-large and Mistral-7B-v0.1)

Interestingly, BERT-large, Mistral-7B-v0.1 and Llama-2-7b-hf all peak at layer 16, which is the same layer reported in (Hewitt and Manning, 2019). At layer 16, the models achieve a LAS on the test set of 70.2, 60.6 and 62.9 respectively. As recently reported in (Eisape et al., 2022) for the Structural Probe, these results suggest that the Polar Probe also works best with Masked Language Models. The Polar Probe, despite its conceptual simplicity matches performance with a more intricate and modular labeled probe (Müller-Eberstein et al., 2022).

**Structural evaluation.**   The Polar Probe is optimized with both a Structural and an Angular objective. This means that the optimization of such probe might affect the original performance of the Structural Probe. To verify that the gap in performance, we compute the UUAS between the

predicted tree and the annotated tree for both the Structural and Polar probe (Fig. 3). The results confirm that the Polar Probe preserves (but does not improve) the syntactic distances between the linear readout of the probed word embeddings.

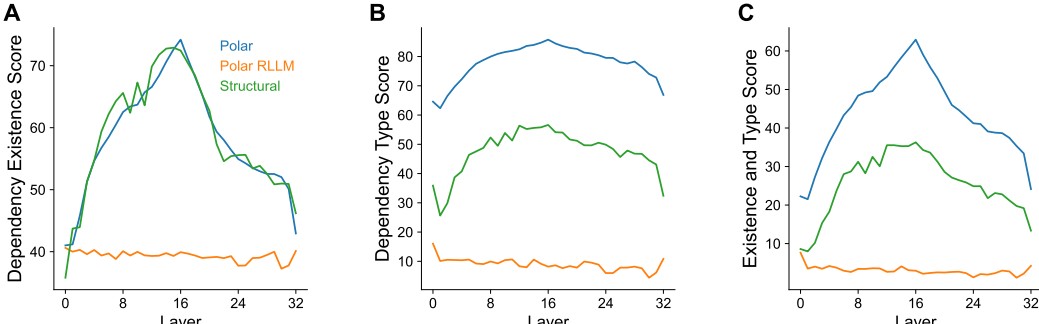

Figure 3: **The Polar Probe outperforms the Structural Probe at identifying labeled and directed dependencies.** **A.** For dependency existence, the Polar Probe matches the UUAS performance of the Structural probe, peaking at layer 16. **B.** For dependency type, the Polar Probe outperforms in Label Accuracy the Structural (LAS) Probe by around 80% across the different layers of Llama-2-7b-hf. **C.** For both dependency existence and type, the Polar Probe outperforms in LAS the Structural Probe by around 90% accross the different layers of Llama-2-7b-hf.

**Dimensionality analysis.** How many dimensions are necessary to successfully represent the full syntactic tree with the proposed polar coordinate system? To address this question, we varied $k''$, namely the dimensionality of the space of the Polar Probe (Fig. 4). Analogously to the rank analysis of Structural Probe (Hewitt and Manning, 2019), we observe a peak around $k'' = 128$. Contrary to theoretical predictions (Smolensky, 1987), these results suggest that the space representing the complete syntactic tree needs not be unreasonably large. For dependency types, this phenomenon could be relatively intuitive, as the unit circle (i.e. only 2 dimensions) can easily separate many different dependency types, such that a weakly non-linear readout would isolate these categories.

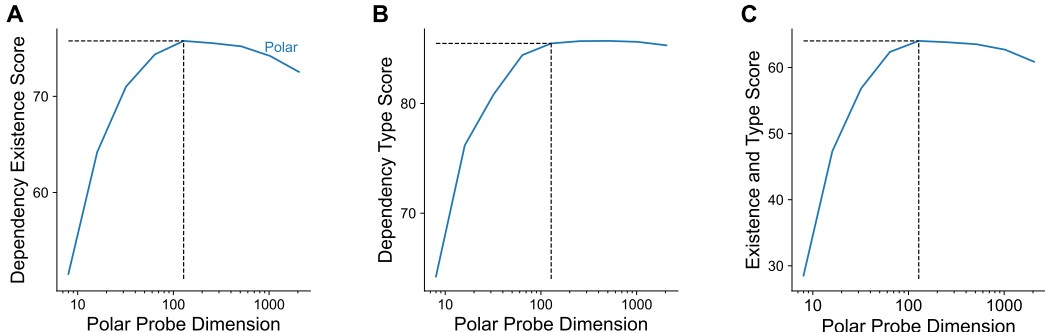

Figure 4: **The optimal dimensionality for the Polar Probe is an order of magnitude small than model's layer size.** Polar Probe performance as a function of dimensionality, measured by **A.** UUAS, **B.** Dependency Type Accuracy and **C.** LAS for Llama-2-7b-hf as a function of $k''$, the dimensionality of the probe's space. The optimal dimensionality for the Polar Probe is 128, achieving the highest LAS.

**Controlled sentences.** Natural sentences are highly variable in structure and content. To verify more precisely the behavior of the Polar Probe, we evaluate it on the "Controlled Dataset" and its different sentence levels: Short, Relative Clause and Long-Nested.

First, we observe that in the space of the Polar Probe, the representation of dependency trees appears to be consistent across sentences' length and substructures. For example, as shown in Fig. 5, the PCA visualization of Polar Probe word embeddings belonging to the main phrase is virtually identical whether it is attached to a relative clause or to a long-nested structure. This invariance supports the notion that dependency trees are represented by a systematic coordinate system that can be recovered with the Polar Probe.

We also observe that dependency types are robustly identified, whether they are part of the main phrase or not.

Overall, these results visually confirm that the Polar Probe reliably represents the dependency structure and dependency types of complex syntactic structures. The latter agrees with the extensive Structural and Angular evaluations performed.

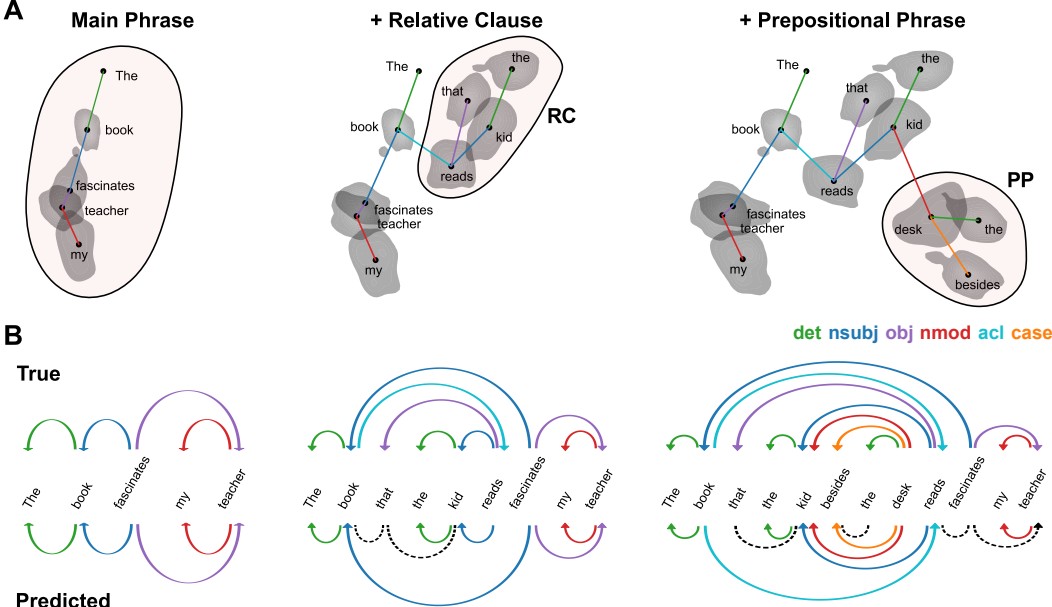

Figure 5: **Visualization of the dependency tree uncovered by the Polar Probe on a set of sentences with increasingly complex hierarchical structures. A.** We display a PCA visualization of the distributions of word embeddings (once linearly read out by the Polar Probe), for the different syntactic levels in the "Controlled Dataset". Each individual distribution corresponds to a specific role of the word in the sentence. The centroids are linked with colored lines, displaying the true syntactic tree of the corresponding sentence. **B.** Most frequent syntactic tree prediction by the Polar Probe for the different syntactic levels. The relations between words are color coded according to the type of syntactic dependency. The incorrectly predicted relations are represented with dashed arrows. That is, either a dependency relation existence (no arrow), or a dependency type (with arrow) was erroneously identified.

## 4 Discussion

**Summary.** We show that within the activation space of language models, there exists a subspace, where syntactic trees are fully represented by a polar coordinate system. There, the presence and type of a syntactic relation between two words is represented by their distance and relative direction, respectively. Importantly, the Polar Probe preserves the structural properties of the Structural Probe (Hewitt and Manning, 2019), but better represents the type of syntactic relations.

**Limitations.** The present work presents four main limitations. First, we only investigate the English language. Yet, human languages use different grammatical rules, and may, consequently, be structured according to different types of trees. Interestingly, as language models become increasingly able to process a wide spectrum of languages (Costa-jussà et al., 2022), the present framework opens the exciting possibility to explore universal (or divergent) grammatical representations in artificial neural networks, following (Müller-Eberstein et al., 2022; Chi et al., 2020).

Second, syntactic structures are not necessarily restricted to the description of relations *between* words. In particular, morphology predicts that words themselves may be represented as trees of morphemes. Consequently, whether and how the present framework generalizes to the different scales of linguistic structures remains to be further investigated.

Third, like the Structural Probe, the Polar Probe is based on a supervised task: we optimize a linear transformation that maximally retrieves a *known* syntactic structure from the neural activations.

Developing an unsupervised probe would be important to help discovers unsuspected syntactic structures. In addition, we here focused on dependency structures. Yet, other formalisms, based on phrase structures (Chomsky, 1957; Joshi and Schabes, 1997; Cinque and Rizzi, 2009; Chomsky, 2014) could offer alternative trees, and could be equally probed through the present framework. This approach could thus offer the possibility of experimentally testing which of these linguistic theories best account for the representations of human languages in neural networks.

Finally, we assume that syntactic trees can be best read out using Euclidean probes. However, alternative assumptions, such as hyperbolic representations, have been a fruitful tool to interpret deep learning models' representations in both text and image modalities (Dhingra et al., 2018; Nickel and Kiela, 2017; Desai et al., 2023). We speculate that this direction could provide a valuable avenue for extending the current work.

## 5  Related work

**Syntax in artificial neural networks.**    Overall, this study complements previous research on syntax in artificial neural networks. Originally, (Smolensky, 1987) demonstrated that vectorial systems could, in principle, represent symbolic structures with tensor products but did not provide an empirical demonstration that neural networks did, in fact, demonstrate this property. More recently, language models were tested on their *capacity* to process syntactic structures by evaluating their behavior on grammatical and ungrammatical sentences (Lakretz et al., 2020, 2021a; Hewitt and Manning, 2019; Hale et al., 2022; Evanson et al., 2023; Linzen et al., 2016). Finally, several groups explored how this capacity was instantiated in the neural activations (Huang et al., 2017; Palangi et al., 2017; Soulos et al., 2019; Lakretz et al., 2019), culminating in (Hewitt and Manning, 2019)'s Structural Probe. Since the discovery of the Structural Probe different adaptations have been developed, notably including hyperbolic (Chen et al., 2021), orthoghonal (Limisiewicz and Mareček, 2021), nonlinear (Eisape et al., 2022; White et al., 2021) variants. The present work completes this long effort by showing how an interpretable syntactic code based on both distances and orientations spontaneously emerges in language models.

**Syntax in biological neural networks.**    This link between linguistics and artificial neural networks holds significant potential for neuroscience. In particular, until the latest rise of large language models, many experimental neuroimaging studies aimed to identify the neural bases of syntax in the human brain (Hale et al., 2022). For example, (Pallier et al., 2011) showed with functional Magnetic Resonance Imaging (fMRI) that several regions of the superior temporal lobe and prefrontal cortex responded proportionally to constituent size. Critically, language models are now becoming standard bases to predict and explain the brain responses to natural language processing: The activations of these artificial neural networks have indeed been shown to linearly map onto fMRI, intracranial and MEG recordings of the brain in responses to the same words and sentences (Jain and Huth, 2018; Caucheteux and King, 2022; Reddy and Wehbe, 2021; Caucheteux et al., 2021; Pasquiou et al., 2022, 2023). However, this mapping remains difficult to interpret, and the neural code for syntax in the brain remains a major unknown. The present work thus provides a testable hypothesis to understand how syntactic trees may be explicitly represented in the brain.

**Broader impact.**    Combined with the works outlined above, our results open the exciting possibility that the polar coordinate system may, in fact, explain how syntax is encoded in the human brain. Critically, this framework may generalize beyond syntactic tree structures, and apply to any compositional problem, including compositional semantics, object-feature binding in vision, and representation of knowledge graphs. Above all, while many have long opposed symbolic and connectionist formalisms, this work contributes to show how these two systems of representations may be largely compatible with one another, as long predicted (Smolensky, 1990; Smolensky et al., 2022). This reconciliation thus holds great promises to understand the brain mechanisms of language and composition.

## 6  Acknowledgments

This project was provided with computer and storage resources by GENCI at IDRIS thanks to the grant 2023-AD011014766 on the supercomputer Jean Zay's the V100 and A100 partition (PDS).

This project has received funding from the European Union's Horizon 2020 research and innovation program under the Marie Sklodowska-Curie grant agreement No 945304 (PDS).

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

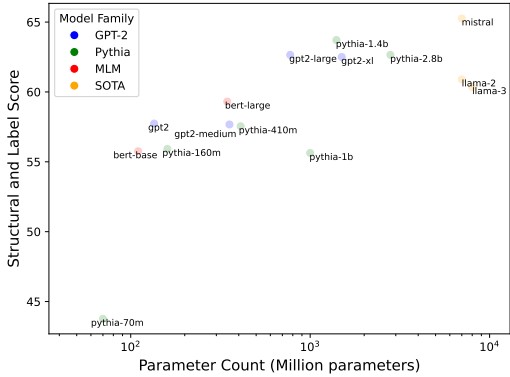

Figure 6: Polar Probe performance on the EN-EWT dataset for Language Models with different families and sizes

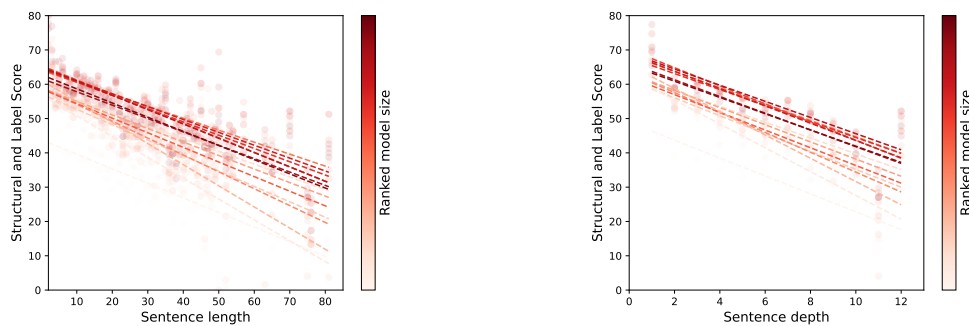

Figure 7: Comparative analysis of Polar Probe performance on the N-EWT dataset as a function of sentence length (left) and sentence depth (right). The scores are shown across various model sizes (ranked by model size), with darker lines indicating larger models.

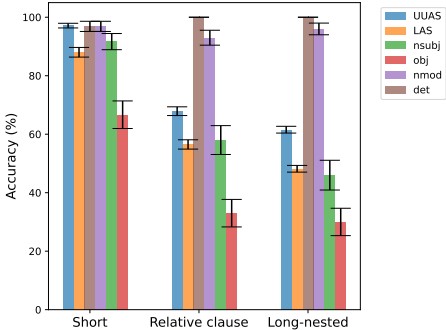

Figure 8: Polar Probe performance across different sentence structures and dependency types in a controlled dataset. The three categories (Short, Relative clause, and Long-nested) show the performance breakdown by Unlabeled Attachment Score (UUAS), Labeled Attachment Score (LAS), and specific dependency relations in the main phrase. Error bars represent the standard error across relations.

