# OpenReview forum: "A Polar coordinate system represents syntax in large language models"
_NeurIPS.cc/2024/Conference — NeurIPS 2024 poster_

### Official Review · Reviewer_rWeD · 2024-07-13

**Soundness:** 4
**Presentation:** 4
**Contribution:** 3
**Rating:** 8
**Confidence:** 4

**Summary:**

This paper proposes polar probes, a kind of structural probe that learns a distance and rotation function that can more accurately classify syntactic structure from language model representations than previous approaches. In particular, the question of whether direction can represent the type of syntactic relationship is answered. The authors find that their polar probe outperforms previous methods quite significantly and show

**Strengths:**

This paper is very well presented and a pleasure to read. The empirical findings are strong and clearly support the hypothesis that the direction, as well as the angle of the representations of an LM projected on a plane represent the syntactic relationships encoded by the model. The authors show that this interpretation is able to much more strongly reconstruct ground truth syntactic parses from hidden state representations than structured probes. The controlled dataset provides a clean comparison in a challenging setting and is a useful resource for future work. A major finding of this work is that it vastly raises the upper bar for how well we should consider syntax to be encoded by language models.

**Weaknesses:**

Weaknesses like the focus on dependency parses and drawbacks of current tokenizers are addressed in limitations, but are still weaknesses nonetheless.

Please include  UUAS, LAS and Balanced Accuracy for the evaluation on the controlled dataset separately for comparison.

As thorough as this paper is, I think it could go deeper on the model analysis. It's nice that the layer-wise analysis is consistent with previous work, but this would be mostly expected. For example, could the authors show that models of different sizes capture more/less syntactic complexity? Is there a critical point where syntax becomes well represented and gains are diminishing after more scaling? Do larger models capture more of the "tail" of rare syntactic constructions? This could be carried out on the GPT2 or Pythia family of models.

Nits:

- please make the text in the legend/axis labels for figure 3 bigger

- Typo L36: "proposed settle"

**Questions:**

N/a

**Limitations:**

Yes

---

> ### Author Rebuttal · Authors · 2024-08-07
>
> We thank reviewer rWeD for their insightful and constructive comments,
>
>
> ### Controlled dataset
>
> We agree that reporting performance metrics such as UUAS, LAS, and Balanced Accuracy on the controlled dataset is an important addition. To address this, we will include a new figure in the appendix of the camera-ready version of the manuscript. This figure will display the UUAS, LAS, and Balanced Accuracy for each of the gold edges in the main phrase, with error bars representing standard error. (The figure is available as Figure 4 in the author rebuttal attached pdf)
> Our result reveals that the degradation in UUAS and LAS is primarily due to the addition of relative clauses and prepositional phrases. Specifically, the accuracy for the ‘nsubj’ relation decreases by approximately 50% after adding these constituents. Unexpectedly, we also observe a degradation of the same magnitude in the ‘obj’ relation, despite the constant distance between words in the ‘obj’ relation across the three sentence levels.
>
>
> ### LAS and model size
>
> In response to the reviewer’s recommendation, we have conducted a study on how LAS varies with model size across different families (Pythia, GPT-2, BERT, and SOTA). The updated figure, which will be added to the camera-ready version, demonstrates a clear trend: larger models generally achieve higher LAS scores. (The figure is available as Figure 1 in the author rebuttal attached pdf)
>
>
> ### LAS and sentence complexity
>
> We also concur that exploring how performance degrades with sentence complexity is valuable. Our new analysis shows how performance varies with sentence length and depth, indicating that larger models are better at capturing the tails of the syntactic complexity distribution. (The new figures are available as Figure 2A and 2B in the author rebuttal attached pdf)
>
> Finally, we will correct the typos and increase the font size in the legends and axis labels of Figure 3 in the camera-ready version.

---

> > ### Comment · Reviewer_rWeD · 2024-08-12
> > **Thank you for the reply. Interesting rebuttal results**
> >
> > Thank you to the authors for their response and additional experiments. I appreciate the work put into the rebuttal to address the finer-grained questions that I think this work raises. I have read the other reviews and reconsidered the paper, and I'm raising my score from a 7 to an 8. Other reviewers raised good points (baselines, Muller-Ebstein), but I agree with the authors responses to these/the authors agree to reasonable proposed changes. I see a lot of value in this approach for model interpretability and testing linguistic hypotheses in models of language. I would like to see this paper published

---

### Official Review · Reviewer_tm4o · 2024-07-13

**Soundness:** 3
**Presentation:** 3
**Contribution:** 2
**Rating:** 5
**Confidence:** 2

**Summary:**

Whereas prior work (Hewitt and Manning 2018) probed syntactic distance and depth, this work proposed to push that forward by also probing headedness and dependency type.  Specifically, this doesn't separately probe those three, but aims for a single vector space where euclidean distance defines syntactic distance but the difference vector maps to a UD dependency label (optimized with a contrastive learning objective).

**Strengths:**

It is a pretty well-written paper, and the framing of the angular probe idea seems well explained and to have some elegance to it (in aiming for a single underlying representation); parts of the implementation seem well-considered to get that single representation.

**Weaknesses:**

- If viewed merely as a combination of probing structure and labeling, it is very similar to a work like Muller-Eberstein et al. 2022.   The advantage of this paper -  having more of a shared representation -- is appealing, but I wish the consequences of that shared space were better explored.
- Analysis was somewhat lacking: for a probing paper, there were relatively work showing what this tells us about the syntactic behavior of models.

**Questions:**

- Have the authors looked at extended dependencies?  The notion of differences as dependency type seems more specific (and to imply more interesting consequences) if there are multiple "paths" through a tree.

**Limitations:**

Yes

---

> ### Author Rebuttal · Authors · 2024-08-07
>
> We thank tm4o for their thorough constructive feedback.
>
> We agree that our discussion of the work by Muller-Eberstein et al. (2022) was not sufficiently detailed, which may have made the novelty of our contributions seem less apparent.
>
> To rectify this, we have expanded our discussion and incorporated four new analyses.
>
> * "Our research shares similarities with Muller-Eberstein et al. (2022), who employed linear probing to classify dependency types in language models (LMs). However, our objectives, method, data and results differ significantly. First, while their study aims to offer 'a lightweight method [to] guide practitioners in selecting an LM encoder for the structured prediction task of dependency parsing,' our goal is to understand how syntactic structures are represented in LMs.
>
> * Second, their method consists of optimizing three separate probes—for structure, labels, and depth—and integrating them with a complex heuristic. In contrast, our method employs contrastive learning to optimize a single probe, trained to identify, in a variety of LMs, a unique subspace that encapsulates all properties of dependency trees.
>
> * Third, the datasets we analyze are not the same. Muller-Eberstein et al. focused on evaluating their probe’s performance on undifferentiated natural sentences. Our study extends beyond this by examining different types of sentences, e.g. synthetic sentences specifically designed to highlight key linguistic phenomena, such as nested structures.
> In summary, our study builds upon prior research (e.g., Hewitt and Manning, Muller-Eberstein 2022) to offer a more unified, compact, and interpretable representation of dependency trees in contemporary language models."
>
>
> To substantiate these elements, we now show, in a revised figure (Figure 3 in the attached pdf), that sentences with similar syntactic structures are represented by the same coordinate system.
>
> “In the space of the Polar Probe, the representation of dependency trees appears to be consistent across sentences’ length and substructures. For example, the coordinates of the main phrase are virtually identical whether it is attached to a prepositional phrase and/or a long-nested structure. This invariance supports the notion that dependency trees are represented by a systematic coordinate system that can be recovered with the Polar Probe.”
>
> Relatedly, we now provide new analyses to clarify the representations of subtrees.
>
> “The analysis of synthetic sentences shows that relative and long-nested substructures are represented, for the most part, less precisely than the main phrases. Similarly, the analysis of natural sentences shows that Label Attachment Scores (LAS) decrease with the length and syntactic depth of the sentence (Figure 2A and B in attached pdf). Overall, these results suggest that the Polar representations of dependency trees lose precision as sentences become increasingly complex. This phenomenon corroborates both with their behavior (Lakkretz et al, Linzen et al) and with human behavior (Silverman & Ratner,  1997).
>
> Finally, to evaluate the universality of this representational system, we now evaluate the Polar Probe across a broader range of models and scenarios.
>
> “LAS tend to increase with the size of the LM, irrespective of its family (e.g. Pythia, GPT, BERT). This gain seems to reflect an overall improvement: Indeed, LAS improves with model size for both short and long sentences, as well as both shallow and deep dependency trees. This result suggests that larger and more recent models progressively shape their representations to fit a Polar Coordinate system.”
>
> ### Questions
>
> Q1:
> Extended Dependencies:
> Our current approach is limited to (valid) binary tree structures. However, the notions of (1) extended dependencies, and (2) multiple paths, (3) ambiguous structures (4) semantic graphs could all, in principle, be investigated through a Polar Coordinates, as this proposal can apply to any (e.g. cyclic) directed and labeled graph. We will amend our discussion to highlight these interesting future research directions.

---

> ### Comment · Reviewer_tm4o · 2024-08-13
>
> Thanks to the authors for their detailed response, and the additional analyses provided on the paper - and so am bumping the score up from 4 to 5. I still feel like there is room for further analysis -- I really like Reviewer's 9Ubp's comment that 'I think the authors were setting up to explore some questions about the meaningfulness in the "hierarchy" of the tree, especially with the controlled sentence dataset, but then I never saw these really come to fruition.'.

---

> ### Author Response · Authors · 2024-08-13
>
> We thank Reviewer tm4o for their response and are grateful that our analyses have been able to address the previous concerns to a satisfactory level.
>
> We acknowledge that the meaningfulness of the Polar Probe's representations on the controlled dataset may not be sufficiently emphasized, as Reviewer 9Ubp noted. To address this issue, we propose adding the following paragraph to the Discussion section:
>
> "One key advantage of the present Polar Probe is its ability to provide linear, compact, and interpretable representation of dependency trees, distinguishing it from other parsing models. Notably, Figure 6 (Figure 3 in the rebuttal PDF) illustrates an unexpected phenomenon: the probe appears to assign the same coordinates to words within identical phrases and syntactic roles, regardless of sentence length, complexity, and semantic variations. This representational invariance offers a promising foundation for further exploration of how recursion and compositionality are effectively represented and processed in neural networks."
>
> We hope this addition clarifies the meaningfulness of hierarchical structures in the Polar Probe and encourages further exploration of its implications for linguistic representation and processing.

---

### Official Review · Reviewer_9UbP · 2024-07-15

**Soundness:** 3
**Presentation:** 3
**Contribution:** 3
**Rating:** 5
**Confidence:** 3

**Summary:**

Previous work introduced linear probes to explore how syntactic relationships are encoded in LLM embeddings.  This work aims to take it a step further and examine how types of syntactic relationships are encoded in the LLMs.  They introduce a polar probe that when optimized can predict the type of syntactic relations via the angle between them.  In a multi-faceted evaluation, the model outperforms baselines (which are essentially ablations of the model) in terms of stronger cosine similarity between the same relations, and in terms of tree accuracy (UAS/LAS).

**Strengths:**

- An interesting paper with a clear contribution, building on existing probing work while asking a couple new research questions

- The results appear convincing

- The potential to explore syntax through the lens of LLMs, especially when LLMs can be easily trained on unlabeled text, or especially when LLMs are increasingly multilingual, points to some exciting future directions.

- The evaluation also includes some linguistically interesting example cases.  Essentially exactly what I would have asked for (in addition to the larger corpora studies)

**Weaknesses:**

- I find the distinction between probing and parsing to be not entirely clear.  At the point where the evaluation is in terms of UAS/LAS, could this not be compared directly to parsers on the same data (especially since building on top of LLM embeddings would be the most performant solution)?  And where would the discrepancies be, and what would that mean?  Do LLMs not encode those relationships?

- In general the paper seems to suffer from a lack of convincing baselines.  The baselines presented -- the structural or angular probe, are steps along the path to the polar probe.

- Cosine similarity between identical syntactic categories is surprisingly low (to me).  The ranking of categories in terms of the strength of that correlation is also surprising, ith things like 'case' being quite strong.  In general there are many "odd" patterns that I don't have an intuitive explanation for why they occur, and aren't discussed in detail in this work.

- There is no dedicated related work.  I do think the parsing literature, and especially the parsing-on-top-of-LLMs literature is relevant.

**Questions:**

Suggestions / Questions:

Q1 - How the trees are predicted is not clear / whether these are a series of independent predictions or whether they are processed sequentially or decoded jointly?

Q2 Do the same syntactic relations that occur at different levels of the sentence have distinct embeddings?  I think the authors were setting up to explore some questions about the meaningfulness in the "hierarchy" of the tree, especially with the controlled sentence dataset, but then I never saw these really come to fruition.  Especially the talk of short/relative/long-nested partitions -- where are these discussed?  Fig. 5 is mentioned (L241) but Fig 5 is fluff.

L33: "and its neuronal substrate."  What?  It's just a model.

L36: "to settle"?  Though the sentence is bizarre regardless

L24-L253, improper citation formats almost everywhere

"According to linguistics, sentences can be described as linear sequences of words connected by a dependency tree", there isn't a "linguistics" -- there are many competing syntactic frameworks and heated debate as to the pros/cons of each framework, but at the end of the day, these are merely formalisms

L83: Squared euclidean distances (between two word embeddings) cannot trivially represent the presence
84 of dependency relations and their types and directions simultaneously.

Why not?  If these are represented in separate subspaces, why is it not possible to represent these three concepts in a vector space?

**Limitations:**

Yes, the limitations are described.

---

> ### Author Rebuttal · Authors · 2024-08-07
>
> We thank reviewer 9UbP for their insightfulreview.
>
> ## Weaknesses
>
> ### Probing vs. Parsing:
>
> We agree that the distinction between probing and parsing is insufficiently clear. We will amend the discussion as follows:
> “The current 'probing' work is related to extensive research on 'parsing'. However, the two fields have distinct objectives, and thus, distinct approaches. Syntactic probes focus on how the vectorial activations of LMs represent the symbolic constructs theorized by linguists (e.g. trees, nodes). For example, the present Polar Probe uses a single linear transformation to provide a compact and interpretable representation of trees through the distance and angle between pairs of word representations. In contrast, parsing studies primarily aim to accurately reconstruct syntactic trees, regardless of the underlying representation. These studies thus prioritize performance over interpretability, which may lead to the adoption of non-linear probes that learn representations that are powerful, but not effectively learnt and utilized by LMs.“
>
> ### Lack of Baselines:
>
> Previous research, such as Hewitt and Manning (2019), provides a baseline included in our figures. We also include an untrained baseline to show that the Polar Probe extracts syntactic knowledge from self-supervised training, not from tokenization or the probing mechanism. To the best of our knowledge, the only other labeled probing work involving LLMs is by Muller-Eberstein (2022). We will add this baseline to the figures of the camera-ready version.
> In addition, we now provide a series of new analyses to evaluate our Polar Probe
> across different LLM families and sizes (attached Figure 1): the results show that the Polar Probe performance improves in more recent and larger models. on variably complex sentences, using the ud-en-ewt test set (attached Figures 2A and 2B): the results show that the Polar Probe gets increasingly imprecise as the length and syntactic depth of the sentences increases.
> on specific substructures (attached Figure 4): the results show that the Polar Probe is less accurate on nested trees than on the main phrase.
>
> These novel analyses clarify the conditions in which the Polar Probe retrieves dependency trees.
>
> ### Cosine Similarities:
>
> This is a good point. We will now add the following clarification to the results section.
>
> “Note that dimensionality impacts cosine similarity: Given that random vectors in a dimension D typically have a cosine similarity of approximately 1/sqrt(D), a cosine similarity of 0.5 can be interpreted as relatively high. Overall, these results support the notion that LLMs effectively represent syntactic relationships. “
>
> ### Pairwise Cosine Similarity Matrix:
> We agree that examining the Polar Probe’s pairwise cosine similarity matrix is valuable, especially to linguistics. In particular, we find that the relations that lead to off-diagonal blocks reflect different categories, which in linguistics, can be subtle. For example, both the case-mark and the aux-cop difference are a common source of confusion among linguists. Such is the case that, for clarity, the following examples are provided on the Universal Dependencies website
>
> The following sentence will be added to L196:
> “We find that higher cosine similarity among off-diagonal blocks in Fig. 2.C reflects subtle linguistic distinctions. For example, distinctions between case-mark relations (case: 'Sue left after the rehearsal' [after - rehearsal] vs. mark: 'Sue left after we did' [after - did]) and aux-cop relations (cop: 'Sue is smart' [is - smart] vs. aux: 'Sue is considered smart' [is - considered]) illustrate these nuanced differences”
>
> ### Related Work:
> Due to space constraints, the Related Work section was omitted in the initial submission. We will add this section in the extra page allowed in the camera-ready version of the paper. A first paragraph can be found at the beginning of our response to reviewer tm4o.
>
> ### Questions:
>
> Q1:
> The trees are predicted following Hewitt and Manning’s approach, adapted to labeled trees. Thus, from the probed euclidean distance matrix, we get a tree by computing the Minimum Spanning Tree (MST) with Kruskal’s algorithm.  Then, the predicted edges are labeled and directed according to the highest absolute cosine similarity with a set of prototypical vectors for each relation type.
> We acknowledge that this procedure is designed for interpretation rather than performance. This approach demonstrates that the differences of performance arise from the nature of the representations rather than the heuristics of the prediction method.
>
> Q2:
> We now revised Figure 5 to clarify its objective (attached Figure 3) and note:
> “In the space of the Polar Probe, the representation of dependency trees appears to be consistent across sentences’ length and substructures. For example, the coordinates of the main phrase are virtually identical whether it is attached to a prepositional phrase and/or a long-nested structure. This invariance supports the notion that dependency trees are represented by a systematic coordinate system that can be recovered with the Polar Probe.”
> Furthermore, we now provide (attached Figure 4) a detailed analysis of the results obtained from the controlled dataset. Overall, these results show that subtrees are less well represented than the main phrase.
>
> L33:
> We agree and will revise it as follows: ”The discrepancy in the system of representations has thus challenged the unification of the theories of human cognitive processes with the underlying computational implementation in the brain.”
>
> Fig1:
> We agree, and will revise it: “According to the dependency grammar framework …”
>
> L83:
> We agree and will revise it::
> “Following the development of the Structural Probe, squared Euclidean distances between probed word embeddings are not designed to represent both the presence of dependency relations and their types and directions simultaneously. ”
> We also correct the typos and citations

---

> > ### Comment · Reviewer_9UbP · 2024-08-13
> > **A marked improvement, but a little thin**
> >
> > I'd like to thank the authors for taking the time to thoroughly answer many of the questions raised in the review.  I think I have a clearer understanding now of how best to interpret some of the results.
> >
> > I also appreciate the additions to the paper, all of which seem like good steps towards rounding out the paper and helping to differentiate it from existing work.
> >
> > I may come back to this, but at this time I believe I am going to keep the score as is, as I think this score is appropriate for the paper, even in light of the suggested improvements.  I am stating that I believe the paper should be accepted even in its current form.  But while Fig 3 in particular seems very promising, it is also hard to ignore that its role in the paper in its current form seems something of an afterthought.  I believe the paper  still struggles a bit to differentiate itself from the previous Muller-Eberstein work (even if the stated goals of each paper are different).  Expanding in the direction of Fig 3 appears to be a very sensible way of accomplishing this, but it seems underexplored in its current state.

---

### Author Rebuttal · Authors · 2024-08-07

We would like to thank the reviewers for the detailed and relevant comments,

## The strengths pointed by the reviewers are:

* The paper is well-written (tm4o, rWeD)
* Clear contribution, the work opens several avenues of research (9UbP, rWeD)
* Convincing results (9UbP, rWeD)
* Paper provides with a controlled dataset with linguistically interesting sentences (9UbP, rWeD)
* Elegant implementation (tm4o)

## The weaknesses pointed by the reviewers are:

* Findings on the controlled dataset are unclear. (tm4o, rWeD )
* Show the universality of the Polar Probe (rWeD)
* Unclear difference with Muller-Eberstein, 2022 (tm4o)
* Lack of baselines (9UbP)



## To address the weaknesses pointed out by the reviewers, we have made four major  modifications:

### Understand better the role of the controlled dataset, (Fig. 5 is not clear)

First, we have now replaced Fig. 5 with a new figure (Fig. 3 in the attached pdf) to clarify the role of the controlled dataset. Our updated results show that sentences are consistently represented in the same coordinates of the probe’s space, irrespective of their length and substructure. This result illustrate the validity of the polar coordinate system proposed in the present study.

Second, we provide new analyses to evaluate the precision (UUAS and LAS) of different substructures of the syntactic tree (Fig. 4 in the attached pdf). The results show that deep structures (preposition phrases, nested phrases) lead to lower performance than the main phrase – A phenomenon which echos with the behavior of both LLMs’ and humans (Silverman,& Ratner, 1997). In addition, the subject-verb agreement performance gets deteriorated as the linear distance in the sentence between both words increases.

Overall, these new elements clarify how syntactic structure can be linearly decoded from LLMs.

### Show the universality of the Polar Probe

We now added new analyses to demonstrate the universality of the Polar Probe representation system across different model families and sizes (Fig. 1 in the attached pdf). Our results show that the Polar Probe's performance is robust across different models and sizes, and that larger models consistently achieve higher LAS performance.

In addition, we also show how the LAS deteriorates with respect to the length and the depth of the sentence. We have added a new figure (Fig. 2 A and B in the attached pdf) showing the better performance of larger models in the tail of the syntactic complexity distribution.
Unclear difference with (Muller-Eberstein, 2022)

We now clarify the differences between our work and (Muller-Eberstein et al. 2022) in terms of goal, method, data, and consequences. Specifically, we have highlighted our goal to understand whether and how syntactic structures are organized in LMs, whereas Muller-Eberstein et al.'s goal is to propose a lightweight method to “guide practitioners in their choice of LM encoder for the structured prediction task of dependency parsing”. In particular, we now clarify in the discussion that our approach optimizes a single and interpretable probe to find a unique subspace that jointly represents the existence, directionality and labels of dependency trees.


### Lack of baselines

To better contextualize the results, we will add the Muller-Eberstein baseline in Figure 3 of the camera-ready version. Additionally, we added extra references with the performance analysis on the controlled dataset and these scores will be a useful resource for the community to better interpret LAS performance. Lastly, we also added an analysis of the performance of the Polar Probe across different model sizes and families (Fig. 1 in the attached pdf).

Overall, these new analyses strengthen our original conclusion.

Once again, we would like to thank our reviewers for their help in improving the study.

---

### Decision · Program_Chairs · 2024-09-25

**Decision:**

Accept (poster)

**Comment:**

Overall, there is a consensus among the reviewers that the paper should be accepted, with a clear contribution and results that support the main idea. There is a need for better baselines and distinguishing the work from previous work in the literature. The authors clarified the difference between their work and previous work (M&E 2023), mentioning that they aim to show that syntax can be represented using polar coordinates, while previous work tried to build encoders for dependency parsing.